# Exploring the Role of CT-Based Delta-Radiomics in Unresectable Vulvar Cancer

**DOI:** 10.3390/diagnostics15232972

**Published:** 2025-11-23

**Authors:** Abdulla Alzibdeh, Bara M. Hammadeh, Rahaf Alnajjar, Mohammad Abd Al-Raheem, Rima Mheidat, Alzahra’a Al Matairi, Mohamed Qamber, Hanan Almasri, Bayan Altalla’, Amal Al-Omari, Fawzi Abuhijla

**Affiliations:** 1King Hussein Cancer Center, Amman 11941, Jordanra.16695@khcc.jo (R.A.); hmasri@khcc.jo (H.A.); ba.16163@khcc.jo (B.A.); asomari@khcc.jo (A.A.-O.); 2Faculty of Medicine, Al-Balqa’ Applied University, Salt 19117, Jordan; barahammadeh2005@gmail.com; 3Faculty of Medicine, University of Jordan, Amman 11942, Jordan

**Keywords:** vulvar cancer, delta-radiomics, radiomics, CT simulation, gross tumor volume (GTV), prognosis and risk stratification, local control, overall survival, adaptive radiotherapy

## Abstract

**Background/Objectives:** To explore the prognostic potential of gross tumor volume (GTV)-based delta-radiomic features from CT simulation scans in patients with locally advanced unresectable vulvar cancer. **Methods:** A total of 21 patients (between 2019 and 2024) undergoing definitive radiotherapy were included, with baseline and post-phase I (after 25 fractions) CT simulation scans analyzed. Radiomic features (*n* = 107) were extracted from GTVs using PyRadiomics, and delta features were calculated as the relative change between scans. A multi-step selection pipeline (univariable Cox screening (*p* < 0.10), correlation filtering, and Lasso–Cox) was applied for each endpoint: local control (LC), regional control, distant metastasis-free survival, progression-free survival, and overall survival (OS). Model discrimination was assessed via 500-iteration bootstrapped concordance index (C-index), and calibration was plotted at 24 months. **Results:** Median follow-up was 50.0 months. The 2-year LC and OS rates were 56.2% and 55.9%, respectively. Final multivariable models retained a sole texture Δ feature for LC (HR = 2.62, 95% CI = 1.05–6.52, *p* = 0.039; C-index = 0.748) and six Δ features for OS (C-index = 0.864). No features were retained for other endpoints. For LC, increased run-length non-uniformity after phase I predicted poorer control. For OS, increased texture/shape complexity predicted worse survival, whereas increased uniformity predicted better survival. **Conclusions:** CT-based delta-radiomic features, particularly shape and texture metrics, may predict LC and OS in unresectable vulvar cancer. Despite the small sample size, these findings highlight the potential for delta-radiomics as a noninvasive biomarker for risk stratification. Validation in larger cohorts and exploring potential in adaptive radiotherapy are warranted.

## 1. Introduction

Vulvar cancer is a rare but increasingly diagnosed gynecological cancer, accounting for nearly 5% of all gynecologic malignancies [1,2]. Squamous cell carcinoma accounts for 95% of cases [3]. It is considered a significant clinical challenge, especially in its advanced stages. The management of locally advanced unresectable vulvar cancer usually involves definitive chemoradiation [4]. However, outcomes remain poor, underscoring the need for improved prognostic tools [5,6,7].

Locoregional rather than distant failures account for most treatment relapses in vulvar cancer, carrying substantial morbidity and offering limited salvage options [8,9]. Distant metastases are uncommon and typically occur only after locoregional relapse [8]. Patients who experience locoregional or distant recurrence have markedly poorer survival rates [10].

Imaging studies play a key role in assessing gynecological cancers. Magnetic resonance imaging (MRI) and ultrasound have established utility in tumor characterization and staging [11,12,13,14]. Nevertheless, CT simulation is universally performed for all patients undergoing radiotherapy, following standardized acquisition protocols and consistent tumor delineation [15]. This routine availability and reproducibility make CT simulation particularly practical for assessing the prognostic significance of radiomic features for vulvar cancer. Radiomics, the high-throughput extraction of quantitative data from medical images, has become a valuable approach for describing tumor heterogeneity and predicting clinical outcomes. Radiomic feature extraction from CT images has become increasingly known as a noninvasive biomarker that has the potential to help in cancer prognostics [16,17]. CT simulation scans are routinely obtained for all radiotherapy patients and thus represent a readily available source of segmented images, with data that can help predict response to treatment, prognosis, and treatment-related toxicities [18,19,20]. In March 2025, Xiao et al. constructed a model combining CT-based radiomic features and clinical indices that can predict overall survival (OS) in cervical cancer patients treated with intensity-modulated radiotherapy (IMRT) and concurrent chemotherapy [21].

Another extension of this work is “delta-radiomics”, which assesses changes in radiomic features over the course of therapy [22]. In cervical cancer, delta-radiomics has shown promising results in predicting intermediate and high-risk pathological factors in patients receiving neoadjuvant therapy [23] and may even outperform the International Federation of Gynecology and Obstetrics (FIGO) stage and MRI-assessed maximum tumor diameter in prognostication of locally advanced cervical cancer treated with chemoradiotherapy [24].

It is noteworthy that radiomics research in vulvar cancer remains limited. To date, only a single study by Collarino et al. has explored radiomic features in vulvar cancer using ^18^F-FDG PET/CT images, focusing on metabolic heterogeneity rather than treatment-induced changes [25]. However, no studies have evaluated CT-based or delta-radiomics approaches in vulvar cancer. Given the ability of delta-radiomics to identify treatment-related changes and predict clinical outcomes, examining its utility in vulvar cancer could provide valuable insights. The present study aims to evaluate the prognostic potential of delta-radiomic features extracted from CT simulation images in patients with locally advanced unresectable vulvar cancer, focusing on their association with local control (LC), regional control (RC), distant metastasis-free survival (DMFS), progression-free survival (PFS), and OS.

## 2. Methods

### 2.1. Patients

In this study, we retrospectively reviewed the records of patients with histologically confirmed, locally advanced, unresectable vulvar squamous cell carcinoma treated at our institution between January 2019 and December 2024. Unresectable disease was defined as primary tumors not amenable to complete surgical excision without unacceptable morbidity, based on multidisciplinary team assessment. Inclusion criteria included (i) availability of baseline CT simulation scans acquired before the start of definitive radiotherapy and a second CT simulation scan acquired after the first phase of treatment (25 fractions), (ii) gross tumor volume (GTV) of the primary vulvar tumor contoured by the treating radiation oncologist on both scans, and (iii) complete clinical and follow-up data for the study endpoints. Patients with incomplete imaging, inadequate ROI segmentation, or missing follow-up information were excluded. A minimum follow-up duration of 3 months was required. Demographic, clinical, and treatment-related variables were retrieved from electronic medical records, including age at diagnosis, performance status, FIGO stage, radiotherapy dose and technique, concurrent chemotherapy administration, and follow-up duration.

### 2.2. Treatment, CT Acquisition Parameters, and Segmentation

All patients received definitive radiotherapy in two phases. Phase I consisted of 45 Gy delivered in twenty-five daily fractions, five fractions per week. The field of radiotherapy included the primary vulvar tumor and inguino-pelvic lymph nodes. Phase II subsequently brought the total dose to 60–66 Gy over 30–33 fractions, with a field encompassing the primary vulvar tumor, and involved lymph nodes were included to the maximum tolerable dose. Concurrent chemotherapy consisted of cisplatin at a dose of 40 mg/mm^2^, administered weekly [26].

All patients underwent CT simulation in the supine position, with straight-leg position. As contrast-enhanced CT is not routinely used in simulation for definitive radiotherapy, all included images were acquired without contrast. Scans were acquired using Philips Brilliance Big Bore CT (85 cm bore) with a 3 mm slice thickness, axial slices, extending from at least the L2 vertebral level to below the perineum. Baseline CT simulation scans were obtained within two weeks prior to the start of radiotherapy, and post-phase I scans were performed after completion of 25 fractions, prior to initiating the second phase of treatment. Contouring and planning were performed using the Pinnacle^3^ 16.2 (Philips Medical System) treatment-planning system [27]. All imaging data were stored in DICOM format and retrieved from the institutional radiotherapy-planning system for subsequent processing and radiomic feature extraction.

### 2.3. Image Preprocessing and Radiomic Workflow Overview

Radiomic features were extracted from CT simulation images acquired before and after the first phase of definitive radiotherapy; the first scan is before starting radiotherapy, and the second scan is after 25 fractions. The pipeline was implemented in Python 3.11.13 (Python Software Foundation, Wilmington, DE, USA; 2024) using PyRadiomics v3.x [28], SimpleITK v2.x, pydicom v2.x, and rt-utils v1.x.

### 2.4. DICOM Image and RT Structure Acquisition

CT simulation data were retrieved in DICOM format. For each patient, the CT series was read using SimpleITK’s ImageSeriesReader, which reconstructs the three-dimensional image volume. The resulting CT image was saved as a NIfTI file. In parallel, the corresponding RT structure set (RTSTRUCT) was accessed using pydicom. The ROI corresponding to the gross tumor volume (GTV) of the primary vulvar tumor contoured by the treating radiation oncologist was then selected for feature extraction.

### 2.5. ROI Mask Generation

The RTSTRUCT file was combined with the CT DICOM series using the RTStructBuilder from the rt_utils package, version 1.2.7. The builder generated a binary mask. Given that the default output shape of the mask was width, height, and slices, the mask array was transposed to ensure correct spatial interpretation by SimpleITK. This transposed binary mask was then converted into a SimpleITK image, and its spatial metadata was aligned with that of the CT image. The resulting ROI mask was saved in NIfTI format.

### 2.6. Radiomic Feature Extraction

Feature extraction was performed using the default PyRadiomics RadiomicsFeatureExtractor with the original image type. Settings included force2D = False, binary mask label = 1, and all feature classes enabled. Default discretization and resampling parameters were used, without any customized configuration files (i.e., no YAML (Yet Another Markup Language) overrides).

The standard PyRadiomics [28] default feature set (*n* = 107) was extracted, consisting of the following:

First-order statistics (18 features).

Three-dimensional shape features (14 features).

Gray-Level Co-occurrence Matrix (GLCM, 24 features).

Gray-Level Run-Length Matrix (GLRLM, 16 features).

Gray-Level Size-Zone Matrix (GLSZM, 16 features).

Gray-Level Dependence Matrix (GLDM, 14 features).

Neighborhood Gray-Tone Difference Matrix (NGTDM, 5 features).

### 2.7. Calculation of Delta-Radiomics

Delta-radiomic features, or Δ features, were computed to capture the intra-treatment variation in imaging features. The formula used was
$$Delta\;Feature=\frac{F_{post}-F_{pre}}{F_{pre}}$$ where
$F_{pre}$ and
$F_{post}$ are the radiomic features from the baseline and post-phase I scans, respectively.

To ensure comparability across features with different scales, a z-score normalization was applied to the Δ features (Δ features). This transformation rescales the data to have a mean of zero and a standard deviation of one.

### 2.8. Survival Endpoints

Survival endpoints were measured from the date of diagnosis to the date of the specified event or last follow-up. Patients known to be event-free were censored at the last date. Time-to-event definitions of each endpoint are as follows:

Local Control (LC): Time until recurrence or persistence of disease within the primary vulvar tumor site as determined by multidisciplinary team assessment based on clinical examination and imaging.

Regional Control (RC): Time until nodal recurrence within the inguinal or pelvic lymph node regions until the common iliac lymph node group.

Distant Metastasis-Free Survival (DMFS): Time until detection of distant metastatic disease outside the pelvis (including distant lymph nodes not included in the RC definition) or death from any cause, whichever occurred first.

Progression-Free Survival (PFS): Time until any disease progression (local, regional, or distant) or death from any cause, whichever occurred first.

Overall Survival (OS): Time until death from any cause.

Note that disease status was determined through multidisciplinary review of clinical notes, imaging (CT, MRI, and/or PET/CT), and pathology reports when available.

### 2.9. Feature Selection for Survival Outcomes

To select delta-radiomic features for prediction of outcomes, we first performed univariable Cox proportional-hazards regression for each Δ feature against each of the five endpoints (LC, RC, DMFS, PFS, and OS), retaining only those with a two-sided *p*-value < 0.10. This liberal screening threshold of *p* < 0.10 was adopted given the small sample size and the exploratory intent of our study to prevent premature exclusion of potentially informative features prior to multivariable analyses. This approach aligns with the prior radiomics and exploratory modeling literature [16,28,29,30].

Next, we computed pairwise Pearson correlations among the retained features and eliminated one member of any pair with |r| > 0.90, dropping the feature with the weaker univariable association. This step reduces redundant features and combats collinearity. The yielded set of Δ features was then fed into a Lasso-penalized Cox model (L1 regularization) with penalty strength λ chosen via five-fold cross-validated concordance index maximization. Δ features with non-zero coefficients at the optimal λ were deemed robust predictors. Finally, Lasso-selected Δ features were entered into an unpenalized Cox model, and a bidirectional stepwise selection based on Akaike’s Information Criterion (AIC) was performed. The complete workflow of image processing and radiomics analysis is illustrated in Figure 1.

### 2.10. Statistical Analysis

Patients’ characteristics are reported as medians, interquartile ranges (IQRs), counts, and percentages. Median follow-up was calculated using the reverse Kaplan–Meier method. Rates of each survival endpoint of LC, RC, DMFS, PFS, and OS are reported as median, 2-year, and 4-year rates. Results of significant univariate Cox proportional-hazards regression are reported in the Appendix A. For multivariable analysis, proportional-hazards assumptions were assessed using Schoenfeld residuals (all global and covariate-specific tests should be *p* > 0.05). For covariates showing evidence of violation, time-varying effects were modeled by adding covariate–log(time) interaction terms, and the stepwise selection procedure was repeated with hierarchical retention of main effects. Multicollinearity was assessed by calculating variance inflation factors (all VIFs should be <5). Results of the final Cox model after feature selection are reported in terms of hazard ratios (HRs) and CI for each Δ feature, along with the 500-iteration bootstrapped C-index. Calibration plots are provided as Appendix A. A *p*-value < 0.05 was deemed significant. Analyses were performed using Python version 3.11.13 (Python Software Foundation, USA; 2024).

## 3. Results

### 3.1. Patient Characteristics

Twenty-one patients with locally advanced unresectable vulvar cancer were included. Table 1 shows patients’ characteristics and treatment specifics. The median age at diagnosis was 57.0 years (range: 37–79 years; IQR: 50.0–67.0). All patients were treated with definitive radiotherapy. Radiotherapy was planned via volumetric modulated arc therapy (VMAT) in all patients. The total vulvar radiotherapy dose had a median of 65.0 Gy (IQR: 64.0–66.6), while the median dose of the first phase of radiotherapy in Gy was 45.0 (IQR: 45.0–50.4). Only one patient did not receive concurrent chemotherapy with radiotherapy. As discussed in Table 1, the median number of chemotherapy cycles was 5 (IQR: 4–6). Three patients underwent inguinal nodal debulking for bulky lymphadenopathy prior to radiotherapy.

Regarding past medical history, 47.6% of patients were medically free otherwise, while the rest of the patients had a plethora of medical diagnoses, as illustrated in Table 1.

### 3.2. Treatment Outcomes

After a median follow-up of 50.0 months (IQR 10.7–59.0), median LC was not reached, with a 2-year rate of 56.2% (95% CI 29.0–76.5). RC median was also not reached, with the 2-year rate being 65.7% (95% CI 33.6–85.1). Median DMFS was 20.0 months (95% CI 11.3–71.0), and the 2-year rate was 44.3% (95% CI 19.3–66.8). Median PFS was 10.0 months (95% CI 8.0–67.0), and the 2-year PFS rate was 36.1% (95% CI 13.7–59.3). Notably, most or all of the events counted by these four endpoints occurred before 2 years, as evident in Table 2. Median OS was 28.0 months (95% CI 16.2–71.0), and the 2-year rate was 55.9% (95% CI 28.4–76.4). Survival curves are illustrated in Figure 2.

### 3.3. Univariable Analyses and Feature Selection

On univariable Cox regression, several significant Δ features were yielded for each endpoint, and those with a *p*-value of <0.1 are reported in Appendix A, along with Kaplan–Meier curves for those with a *p*-value of <0.5 (Appendix A; none had a *p*-value of <0.5 for RC).

All Δ features with a *p*-value of <0.1 were included in the pairwise Pearson correlations (Appendix A). After excluding redundant Δ features, a Lasso-penalized Cox model was run for each endpoint. At this step, only LC and OS retained any Δ features; no features survived selection for RC, DMFS, or PFS. Selected Δ features that were included in the multivariable analyses are illustrated in Table 3. Appendix A shows the temporal changes in Δ features retained in final models.

### 3.4. Multivariable Analyses and Internal Validation

After AIC-based bidirectional stepwise selection, the final multivariable Cox model for LC only retained the Δ RLNU_norm with a coefficient of 0.962 (HR: 2.62, 95% CI: 1.05–6.52; *p* = 0.039, Table 4). This translates into more than doubling of the hazard of local recurrence for each one standard deviation increase in Δ RLNU_norm. The 500-iteration bootstrap yielded a mean C-index of 0.748 (95% CI 0.506–0.987, Figure 3).

For OS, time-varying effects were incorporated in the final multivariable Cox proportional-hazards model to address violations of the proportional-hazards assumption identified in initial testing. Bidirectional stepwise selection retained six Δ features (Table 4). Among these, Δ DiffAvg, Δ SVR, Δ DiffVar, and Δ GLNU_norm were statistically significant (*p* < 0.05); greater increases in texture or contrast (DiffVar, SVR) seem to portend worse survival, whereas larger increases in uniformity metrics (DiffAvg, GLNU_norm) predict better survival. All retained variables satisfied the proportional-hazards assumption in the final model. The bootstrap-corrected C-index for the OS model was 0.864 (95% CI 0.633–0.973, Figure 3). Calibration plots are illustrated in Appendix A.

## 4. Discussion

This hypothesis-generating study is the first to provide proof-of-concept that CT GTV-based delta-radiomics may serve as a noninvasive biomarker in patients with locally advanced unresectable vulvar cancer. Using a rigorous feature-selection pipeline, including univariable Cox screening, correlation filtering, Cox–Lasso regularization, and AIC-based selection, several features were nominated to predict LC and OS, with no features retained for RC, DMFS, and PFS. This likely reflects the source of the features, the GTV, and highlights the fact that predictors of the more common failure pattern, local failure, are likely to predict worse OS in our group of patients.

Such dimensionality reduction is of paramount importance in radiomics studies, where multiple testing can lead to falsely significant findings [31]. Although multiple significant Δ features on univariable Cox analyses were yielded (Appendix A), none were retained after our multi-step feature-selection process, as described in the results.

Our multivariable Cox model for LC identified one significant feature, RLNU_norm. This texture-based metric captures how uniform the run lengths (i.e., sequences of the same gray level) are in the GTV [28]. Each one-unit increase in the RLNU_norm is associated with a 2.6-fold higher hazard of local recurrence (Table 4); the more run-length heterogeneity the tumor has after the first phase of radiotherapy, the worse its LC is expected to be. As higher RLNU_norm values reflect greater intratumoral texture irregularity, such a feature may reflect hypoxic or radio-resistant tumor subregions [16].

For OS, the model retained six Δ features, of which four were significant, indicating that increased uniformity (e.g., in Δ GLNU_norm and Δ DiffAvg) predicted better survival, while increased heterogeneity or complexity (Δ SVR and Δ DiffVar) predicted worse outcomes.

Our multivariable model for OS had a 500-iteration bootstrap C-index of 0.87 (95% CI 0.61–0.97), indicating a strong discrimination, despite the limited sample size and only nine captured events. Such findings suggest a promising role of radiomics and delta-radiomics for vulvar cancer prognostication and risk stratification.

It is noteworthy that minor time-dependent effects were observed for some Δ-features in the OS multivariable analysis. To account for these, time-varying coefficients were utilized by including covariate–log(time) interaction terms. This adjustment improved model calibration without materially altering the direction or significance of the affected predictors. Although isolated proportionality deviations were observed, their magnitude was small, and their correction did not change the overall prognostic conclusions.

Our findings suggest that residual tumors exhibiting more irregular shapes and higher texture heterogeneity during radiotherapy may reflect treatment resistance and portend worse outcomes. Increasing SVR, a shape feature, reflects the increasing tumor surface complexity relative to volume during treatment. This change can be attributed to the development of infiltrative growth patterns, spiculated borders resulting from radiation-induced fibrosis (or the persistence of aggressive tumor clones), and a necrotic core that leaves uneven tumor surfaces [32].

It has been shown that in carcinoma, metastatic potential is acquired through early molecular changes such as epithelial-to-mesenchymal transformation, orchestrated by the tumor microenvironment and stromal interactions [33]. Whether this heterogeneity in radiomic features reflects the molecular diversity of tumor clones remains a question for future translational studies. Another mystery to be unfolded is whether using subregional segmentation instead of the whole GTV as the region of interest to extract radiomic features would capture the well-known intratumoral heterogeneity, which may help sharpen the performance of radiomics [34].

We chose to focus on unresectable vulvar cancer because prognostic markers are less well established in this setting. In such cases, tumor-derived radiomic features may represent an important source of prognostic information, whereas in post-surgical scenarios, numerous other determinants, such as pathological findings, margin status, and treatment-related factors, are readily available and have been validated to play a substantial role in outcome prediction [35].

Prior delta-radiomic studies in cervical cancer have demonstrated prognostic value for response to treatment and survival outcomes [21,23,24]. However, vulvar cancer has been underrepresented in radiomics research, with a single study that used ^18^F-FDG PET/CT-based radiomics to correlate with tumor biology and predict prognosis [25]. Our study addresses this gap and suggests that even in rare cancers, quantitative imaging biomarkers can be developed using standardized pipelines. Moreover, our findings align with prior evidence suggesting that local control, more than distant failure, remains the main challenge in locally advanced vulvar cancer and potentially determines outcomes in vulvar cancer [7,10,36], thus justifying the focus on GTV-derived features.

CT simulation scans are routinely obtained in radiotherapy planning, and our reliance on these widely available images enhances clinical applicability. CT simulation is easily obtained before and during radiotherapy, with standardized acquisition protocols. Such scans are contoured by experienced radiation oncologists according to established contouring guidelines [15]. Thus, using CT simulation to extract radiomic features ensures reproducibility and facilitates consistent feature extraction across patients and institutions. While MRI and PET/CT may offer more detailed soft-tissue characterization or metabolic data [24,25], their availability, cost, and variability in segmentation limit generalizability. Furthermore, repeated MRI or PET/CT scan acquisitions during radiotherapy are not routinely feasible due to constraints of availability, cost, and patient access, particularly in resource-limited settings. In contrast, CT simulation is already embedded in clinical practice. This ensures easier applicability of delta-radiomics in real-world radiotherapy and facilitates potential integration in adaptive radiotherapy settings. Nevertheless, future research incorporating MRI-based or multimodality radiomics remains warranted.

A model based solely on Δ features from segmented CT scans could be integrated into existing workflows with minimal disruption across different socioeconomic settings. Moreover, results on delta-radiomic features raise the potential for adaptive radiotherapy, where patients with unfavorable delta-radiomic profiles could be flagged early for treatment modifications [37,38]. In fact, prior clinical observations in vulvar cancer indicate that more than half of patients require adaptive replanning during definitive radiotherapy, particularly those with large baseline tumor volumes [39].

### Limitations

Our study has several limitations. First, its single-center, retrospective design, small sample size, and lack of external validation reduce its statistical power and increase the risk of overfitting, all of which may limit the generalizability and reproducibility of the findings. Also, the small sample size hindered the calculation of distant control analysis, as some studies report this event in as few as 5.1% of the cases only [8]. Nonetheless, our limited sample size reflects the rarity of unresectable vulvar cancer. Such an obstacle was mitigated by using a multi-step feature-selection pipeline with internal bootstrapping and calibration to ensure robustness and reduce overfitting. Accordingly, our results should be interpreted as exploratory, but they provide a rationale for larger multi-institutional validation. Future multi-institutional collaboration will be essential for external validation.

In multivariable analyses, stepwise selection using AIC was utilized as it balances model goodness-of-fit and parsimony by penalizing excess parameters [40]. While stepwise selection can predispose to overfitting, this risk was mitigated through prior LASSO regularization, five-fold cross-validation, and 500-iteration bootstrap validation of concordance indices [41]. Nonetheless, some degree of overfitting remains possible given the limited sample size.

Another limitation is the potential subjectivity in ROI contouring, which may affect reproducibility. This reflects the complexity of available consensuses for vulvar cancer contouring, which are difficult to be prescriptive for every patient scenario, especially in the locally advanced unresectable setting [15,42]. In our study, contouring variability was minimized by adhering to established institutional contouring guidelines for vulvar cancer, which align with the available consensuses [15]. For each patient, the same radiation oncology consultant delineated the GTV on both the baseline and mid-treatment scans. Nonetheless, we acknowledge the limitation of single-observer bias and recommend that future studies incorporate multi-observer contouring with reproducibility assessments (e.g., Dice similarity coefficient, intraclass correlation).

FIGO staging was not included in the final model to avoid overloading the analysis with predictors, given the small sample size. Moreover, staging variability was minimal in our cohort, as nearly all patients had stage III–IV disease.

Although contrast-enhanced CT could potentially enhance the prognostic performance of radiomic models, it is not routinely used in simulation for vulvar cancer radiotherapy. Also, feature extraction from contrast-enhanced CT scans introduces potential variability and may not be reproducible across institutions. Using non-enhanced images is more translatable to standard clinical workflows and is safer in patients with kidney impairment.

## 5. Conclusions

In this study, we demonstrate that CT-based delta-radiomic features, particularly shape and texture metrics, hold a potential prognostic value in locally advanced unresectable vulvar cancer. These features capture intra-treatment changes in tumor morphology and heterogeneity that correlate with LC and OS, potentially offering earlier and more objective risk stratification than conventional methods. While our single-center experience provides foundational evidence for delta-radiomics in this rare malignancy, larger multi-institutional studies are needed to validate these findings. Such noninvasive biomarkers provide hope to guide personalized therapeutic strategies. Future work should integrate radiomics with histopathological and genomic data to elucidate the biological underpinnings of these imaging phenotypes.

## Figures and Tables

**Figure 1 diagnostics-15-02972-f001:**
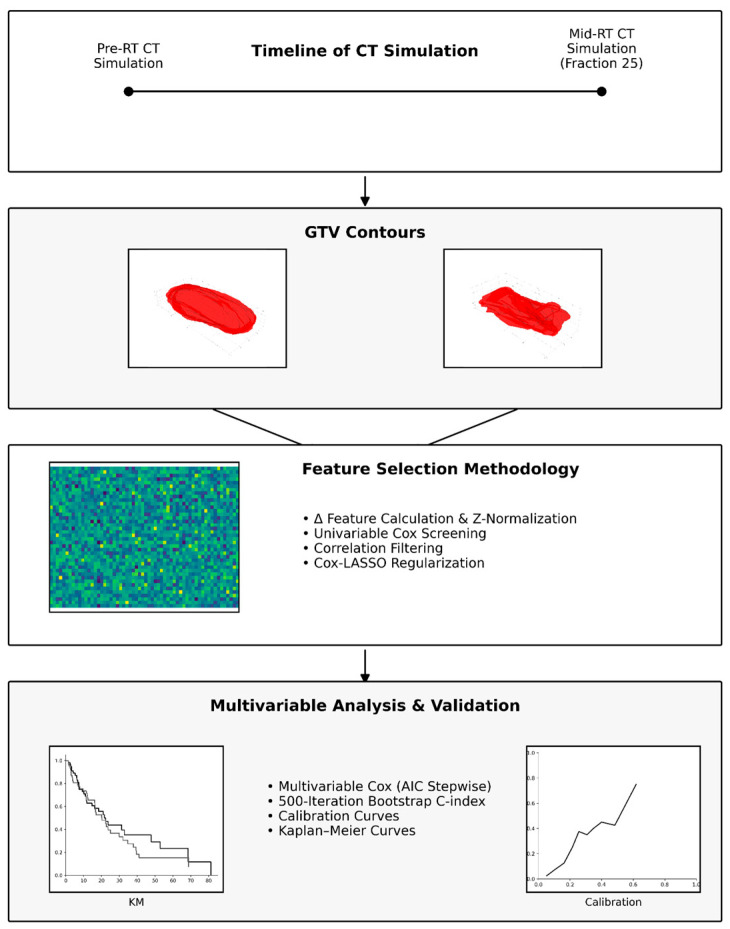
Radiomics analysis workflow. Radiomic features were extracted from gross tumor volume (GTV) masks delineated on pre- and mid-radiotherapy CT simulation scans. Delta features (Δ features) were calculated and Z-normalized. Feature selection was performed sequentially using univariable Cox proportional-hazards screening, correlation filtering, and Cox–LASSO regularization to identify predictors of local control (LC), regional control (RC), distant metastasis-free survival (DMFS), progression-free survival (PFS), and overall survival (OS). Several features were retained for LC and OS prediction, with no features retained for RC, DMFS, or PFS. Final models were developed using multivariable Cox regression with Akaike Information Criterion (AIC)-based stepwise selection, and their performance was evaluated using 500-iteration bootstrap concordance index (C-index) estimation, calibration curves, and Kaplan–Meier survival analysis.

**Figure 2 diagnostics-15-02972-f002:**
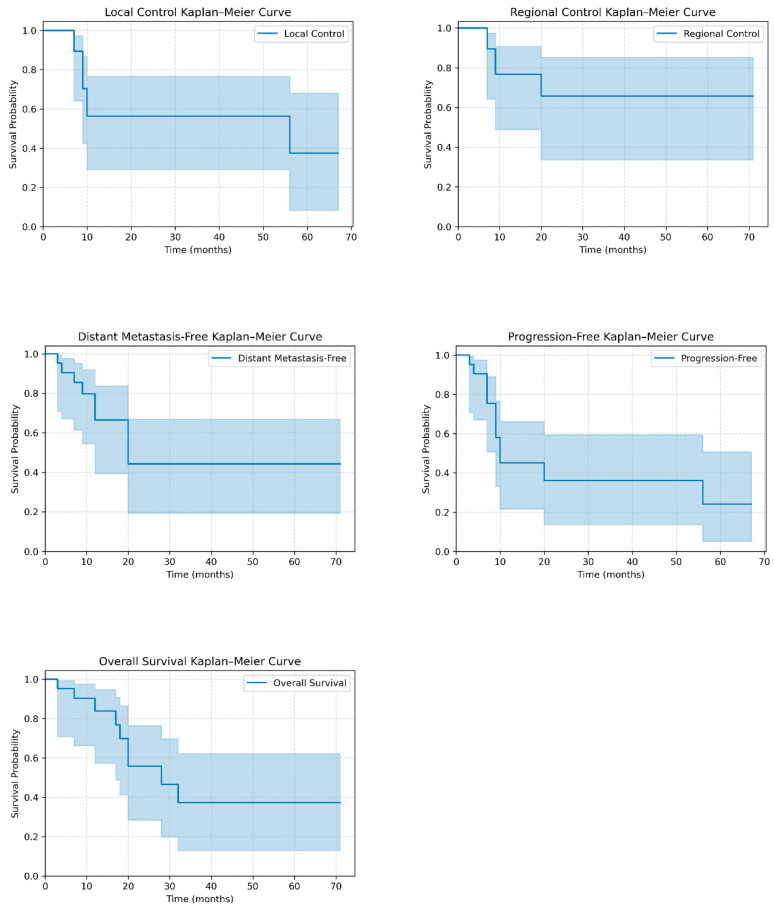
Kaplan–Meier curves for local control (LC), regional control (RC), distant metastasis-free survival (DMFS), progression-free survival (PFS), and overall survival (OS) in the study cohort.

**Figure 3 diagnostics-15-02972-f003:**
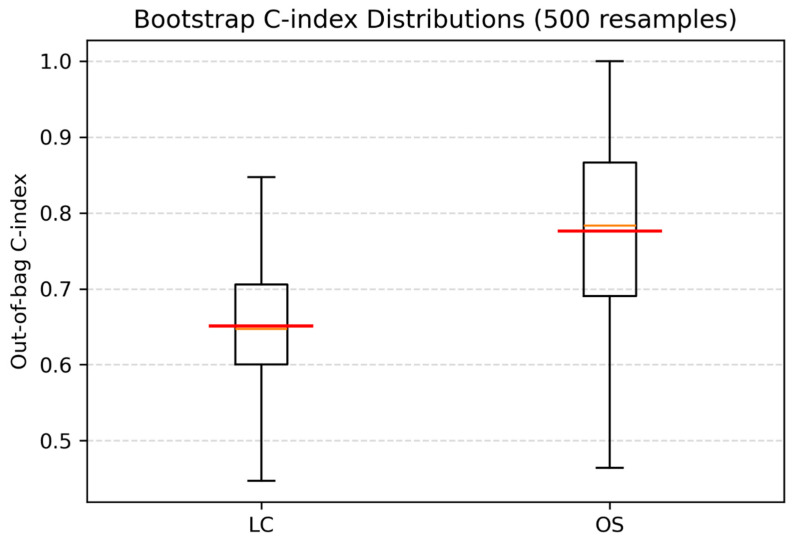
Bootstrapped concordance index (C-index) for the final multivariable Cox models predicting local control (LC) and overall survival (OS). The red line represents the mean C-index, and the yellow line indicates the median C-index across resamples. The LC model retained only ΔRLNU_norm (C-index = 0.751), while the OS model retained six Δ features (C-index = 0.896).

**Table 1 diagnostics-15-02972-t001:** Patients and disease characteristics and treatment specifics (*n* = 21).

Variable	Median		IQR
**Age at diagnosis (years)**	57		50.0–67.0
**Total vulvar dose (Gy)**	65		64.0–66.6
**First-phase external-beam radiotherapy dose in Gy**	45		45.0–50.4
**External-beam radiotherapy number of fractions**	25		25.0–28.0
**Nodal dose (Gy)**	63		54.0–66.0
**EBRT time (days)**	54		51.0–61.0
**Chemotherapy cycles**	5		4.0–5.0
**Variable**	**Category**	**Count**	**Percentage (%)**
**Chronic comorbidities**	Diabetes mellitus	5	23.8
	Hypertension	7	33.3
	Ischemic heart disease	2	9.5
	Chronic obstructive pulmonary disease	1	4.8
	Systemic lupus erythematosus	1	4.8
	Medically free otherwise	10	47.6
**Grade**	1	1	4.8
	2	15	71.4
	3	4	19
	Missing	1	4.8
**HPV categories**	Not associated	12	57.1
	HPV-associated	7	33.3
	Missing	2	9.5
**FIGO stage**	II	1	4.8
	IIIA	1	4.8
	IIIB	7	33.3
	IIIC	1	4.8
	IVA	2	9.5
	IVB	6	28.6
	Missing	3	14.3
**Concurrent chemotherapy**	Yes	20	95.2
	No	1	4.8
**Nodal debulking surgery prior to radiotherapy**	Yes	3	14.3
	No	18	85.7

**Table 2 diagnostics-15-02972-t002:** Treatment outcomes.

Endpoint	Events (*n*)	Total (*n*)	Event Rate (%)	Events ≤ 24 Months (*n*)	≤24 Months (%)
**Local Control**	8	21	38.1	7	87.5
**Regional Control**	5	21	23.8	5	100.0
**Distant Metastasis-Free**	9	21	42.9	9	100.0
**Progression-Free**	12	21	57.1	11	91.7
**Overall Survival**	9	21	42.9	7	77.8

**Table 3 diagnostics-15-02972-t003:** Feature selection results for study endpoints.

Endpoint	Selected Δ Features
**L**C	GLCM Inverse Difference Moment (IDM)
	GLRLM Run Length Non-Uniformity Normalized (RLNU_norm)
	GLRLM Run Percentage (RP)
	First-Order Entropy
	GLCM Difference Entropy (DiffEnt)
	GLCM Cluster Prominence (ClusProm)
RC	*none retained*
**DMFS**	*none retained*
**PFS**	*none retained*
**O**S	GLCM Difference Average (DiffAvg)
	Shape Surface–Volume Ratio (SVR)
	GLCM Difference Variance (DiffVar)
	GLDM Large Dependence Low Gray-Level Emphasis (LDLGLE)
	GLSZM Size-Zone Non-Uniformity (SZNU)
	GLSZM Gray-Level Non-Uniformity Normalized (GLNU_norm)
	GLSZM Zone Entropy (ZoneEnt)
	GLSZM Gray-Level Variance (GLVar)
	First-Order Energy

LC, Local Control; RC, Regional Control; DMFS, Distant Metastasis-Free Survival; PFS, Progression-Free Survival; OS, Overall Survival; GLCM, Gray-Level Co-occurrence Matrix; GLRLM, Gray-Level Run-Length Matrix; GLSZM, Gray-Level Size-Zone Matrix; GLDM, Gray-Level Dependence Matrix; First-order features are derived from the voxel intensity histogram without spatial context.

**Table 4 diagnostics-15-02972-t004:** Multivariable Cox model with AIC-based bidirectional stepwise selection for LC and OS.

Multivariable Cox Model for LC (Retaining One Variable)				
Δ Feature	Coef	HR	95% CI	*p*-Value *
GLRLM Run-Length Non-Uniformity Normalized (**RLNU_norm**)	0.9625	2.618	[1.05, 6.52]	0.0388
**Multivariable Cox Model for OS (time-varying)**				
Feature	Coef	HR	95% CI	*p*-value *
GLCM Difference Average (DiffAvg)	−9.0097	0.00012	[3 × 10^−8^, 0.48]	0.0327
Shape Surface–Volume Ratio (SVR)	5.7666	319.45	[1.74, 5.9 × 10^4^]	0.0302
GLCM Difference Variance (DiffVar)	5.7931	328.02	[1.33, 8.1 × 10^4^]	0.0393
GLDM Large Dependence Low Gray-Level Emphasis (LDLGLE)	−8.3732	0.00023	[1 × 10^−8^, 5.28]	0.1021
GLSZM Gray-Level Non-Uniformity Normalized (GLNU_norm)	−9.2533	0.00010	[3 × 10^−8^, 0.33]	0.0259
First-Order Energy	−1.6158	0.1987	[0.03, 1.22]	0.0807

LC, Local Control; OS, Overall Survival; GLCM, Gray-Level Co-occurrence Matrix; GLRLM, Gray-Level Run-Length Matrix; GLSZM, Gray-Level Size-Zone Matrix; GLDM, Gray-Level Dependence Matrix; First-order features are derived from the voxel intensity histogram without spatial context. * Statistical test: Multivariable Cox proportional-hazards regression.

## Data Availability

The original contributions presented in this study are included in the article/Appendix A. Further inquiries can be directed to the corresponding author.

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
