# Peer review of "Exploring the Role of CT-Based Delta-Radiomics in Unresectable Vulvar Cancer"

_diagnostics, 2025, doi:10.3390/diagnostics15232972_

Round 1
Reviewer 1 Report
Comments and Suggestions for Authors
Thank your for your paper. I have some concerns:
The study should mention why the CT-based Delta-Radiomics is necessary instead of MRI.
Role of ultrasound and MRI in assessing gynecologic tumor should be mentioned. doi: 10.1002/jcu.24046.
A study flowchart with inclusion and exclusion cases should be mentioned.
Sample size was not calculated.
Outcome variables should be described in detailed.
Chemotherapy and surgery should be described more.
Type of HPV should be mentioned.
Value median (IQR) should be presented as median IQR [Q1-Q3] (25%-75%).
Statistical tests should be added to P-value in tables.
Year and country of product should be given to Python version 3.11.13.
Full-words should add to abbreviation word when it first appears. For example: FIGO
Author Response
Comment 1: The study should mention why CT-based delta-radiomics is necessary instead of MRI.
Response: In the revised manuscript, sections “introduction” and “discussion”, we emphasize that as CT simulation scans are universally available in radiotherapy workflows, and the region of interest (ROI), which is the GTV here, is contoured by a radiation oncologist based on available guidelines and body of data, features of such images were used in our study. CT provides a reproducible, standardized imaging source, allowing greater clinical translation across diverse settings. We believe MRI features would likely also have clinical relevance, and are worthy of investigation in future studies, yet MRI is not always feasible, especially doing MRI twice during radiotherapy to evaluate delta radiomic features, due to cost, access, or other reasons.
Comment 2: Role of ultrasound and MRI in assessing gynecologic tumors should be mentioned (doi: 10.1002/jcu.24046).
Response: Thank you for referring this study. This and our study provide a wave of literature to highlight the treasure that is imaging features in sharpening our view of gynecological cancers. We have now cited the suggested reference and highlighted the roles of MRI and ultrasound in tumor characterization. It is noteworthy, however, that CT simulation is uniquely available for all patients before and during radiotherapy, with standardized protocols of image obtainment and GTV segmentation, and hence its practicality for delta-radiomics exceeds that of MRI and ultrasound.
Comment 3: A study flowchart with inclusion and exclusion cases should be mentioned.
Response: A study flowchart illustrating patient inclusion and exclusion is now added as Supplementary Figure 1, and the numbers of supplementary figures were accordingly re-numbered.
Between January 2019 and December 2024, 31 patients with histologically confirmed, locally advanced unresectable vulvar squamous cell carcinoma received definitive radiotherapy at our institution. Of these, 10 patients were excluded: one due to incomplete treatment (death during therapy), one due to insufficient follow-up (<3 months), and eight due to absence of post–phase-I CT simulation (treated with a single radiation plan based only on baseline CT). The final cohort included 21 patients, all of whom met eligibility criteria for delta-radiomics analysis (you can refer to Supplementary Figure 1).
Comment 4: Sample size was not calculated.
Response: No prospective sample size calculation was performed given the retrospective nature of the study and rarity of the disease. Instead, all eligible patients during 2019–2024 were included. As mentioned earlier, a study flowchart illustrating patient inclusion and exclusion is now added as Supplementary Figure 1.
Comment 5: Outcome variables should be described in detail.
Response: We expanded the “Methods” section to define each outcome (LC, RC, DMFS, PFS, OS), including time-to-event definitions and censoring.
Comment 6: Chemotherapy and surgery should be described more.
Response: details on concurrent chemotherapy (weekly cisplatin, number of cycles, dose adjustments) and about surgery are now included in the “Results”. All patients were planned for concurrent chemotherapy, consisting of cisplatin at dose of 40mg/mm2, administered weekly. As discussed in Table 1, median number of chemotherapy cycles was 5 (IQR: 4-6). Surgical history: Our study included patients with unresectable disease only, defined as primary tumors not amenable to complete surgical excision without unacceptable morbidity, based on multidisciplinary team assessment. Any patient how underwent any resection to the primary vulvar tumor was excluded from our study, as the ROI of our radiomic features (the GTV of primary vulvar tumor) would be manipulated. For nodal disease, three patients underwent inguinal nodal debulking for bulky lymphadenopathy prior to radiotherapy; this was included in Table 1 now.
Comment 7: Type of HPV should be mentioned.
Response: HPV status is reported in Table 1. We noted that 33.3% were HPV-associated, 57.1% non-associated, and 2 cases missing. At our center, the HPV status is based on P16 immunostaining, and therefore the specific HPV type (16, 18, etc) are not determined in our pathology samples.
Comment 8: Median (IQR) should be presented as median [Q1–Q3].
Response: We reformatted all median/IQR values accordingly.
Comment 9: Statistical tests should be added to P-values in tables.
Response: The statistical tests used in the tables are as follows (manuscript edited accordingly):
- Table 4: Multivariable Cox proportional‐hazards regression
- Supplementary table 1: Univariable Cox proportional‐hazards regression
- Supplementary Table 3: Wilcoxon signed-rank test
Comment 10: Year and country of product should be given to Python version 3.11.13.
Response: Analyses were done using Python 3.11.13 (conda-forge distribution, 2024 build) within a Jupyter Notebook environment (version 7.3.2) on a 64-bit Windows system. This is now specified in Methods: “Analyses were performed using Python version 3.11.13 (Python Software Foundation, USA; 2024).”
Comment 11: Full words should be added to abbreviation at first appearance (e.g., FIGO).
Response: Introduction using full words was done for all abbreviations at first appearance in edited manuscript (“International Federation of Gynecology and Obstetrics” for FIGO).
Reviewer 2 Report
Comments and Suggestions for Authors
Dear Authors,
This is a study of 21 patients the prognostic role of GTV-based delta-radiomics in advanced unresectable vulvar cancer. This is done on the lines of its prognostic role in cervical Cancer. It's easy to do, and the patients with unfavourable delta-radiomic profiles could be flagged early for treatment modification
The novelty and comprehensive statistical analysis have made it an interesting read. I accept the manuscript in its present form
Author Response
Overall: The reviewer finds the manuscript novel, statistically sound, and accepts it in the current form.
Response: Thank you for this encouraging assessment and for taking the time to review our project.
Reviewer 3 Report
Comments and Suggestions for Authors
Dear authors,
The rationale for using delta-radiomics is well-presented. It's clear why this approach is promising for understanding treatment response.
The introduction could benefit from a more thorough review of existing radiomics literature in vulvar cancer (or the lack thereof). Please emphasize the gap in knowledge that this study aims to address.
Methods:
- The inclusion and exclusion criteria are well-defined. However, it would be helpful to provide more details on the characteristics of the excluded patients and the reasons for their exclusion.
CT Acquisition and Segmentation:
- The lack of contrast enhancement in CT scans is noted. Discuss the potential impact on feature extraction and consider whether this could introduce bias. This is crucial!
- I consider the dependence on a single radiation oncologist for GTV contouring introduces potential subjectivity… Consider discussing inter-observer variability and whether any measures were taken to mitigate this.
Radiomic Feature Extraction:
- Clearly state the software versions and specific parameter settings used for feature extraction. This is important for reproducibility.
- Explain the rationale for selecting the specific set of 107 radiomic features.
Feature Selection and Statistical Analysis:
- The multi-step feature selection pipeline is rigorous. However, the reliance on univariate Cox screening with a relatively liberal p-value threshold (p < 0.10) could lead to the inclusion of spurious associations. Please justify this choice.
- Address Proportional Hazards Assumption Violations: Describe the extent to which these violations were present and their potential impact on the results.
Results:
- Provide a more detailed description of the patient cohort, including comorbidities, prior treatments, and other relevant clinical factors.
- The confidence intervals for the survival rates are wide, reflecting the small sample size... Acknowledge this and temper the interpretation of these results!
- Clearly present the results of the univariate Cox analyses (perhaps in a supplementary table).
- Justify the use of AIC-based stepwise selection. Discuss its limitations and potential for overfitting.
- Carefully interpret the clinical significance of the identified radiomic features, considering their biological plausibility and potential for clinical translation.
- For OS, address the Time-Varying Covariates by incorporating time-varying effects (it is important).
Discussion:
- Explicitly discuss the potential for selection bias, given the retrospective study design!!!
- Acknowledge the lack of external validation and the implications for the generalizability of the findings.
- Address the potential subjectivity in ROI contouring and its impact on reproducibility.
- Provide a more thorough comparison of the findings with other studies in vulvar cancer and other gynecological cancers.
- Discuss the potential advantages and disadvantages of using CT-based radiomics compared to other imaging modalities like MRI or PET/CT.
Also, emphasize the need for multi-institutional validation studies with larger patient cohorts.
Author Response
Comment 1: Expand introduction with more thorough review of radiomics literature in vulvar cancer and emphasize the knowledge gap.
Response: We expanded our introduction to highlight the scarcity of data on radiomics in vulvar cancer prognostication, and discussed the only prior PET/CT radiomics study in vulvar cancer (Collarino et al., J Nucl Med 2019) and emphasized the lack of CT- or MRI-based delta-radiomics studies in this disease.
Comment 2: Provide more details on excluded patients.
Response: the edited manuscript now includes a supplementary figure to show a flowchart of patient inclusion and exclusion. Between 2019 and 2024, 31 patients with histologically confirmed locally advanced unresectable vulvar squamous cell carcinoma were assessed for eligibility. Ten patients were excluded—one due to incomplete treatment (death during therapy), one with follow-up < 3 months, and eight without a post-phase-I CT simulation scan—leaving 21 patients for the final delta-radiomics analysis.
Comment 3: Discuss impact of non-contrast CT and inter-observer variability.
Response: As we addressed in the “Discussion”, contrast-enhanced CT may improve lesion conspicuity and could enhance the prognostic performance of radiomic models, but contrast is not routinely used in CT simulation for vulvar cancer radiotherapy. Also, variation in contrast timing, dose, and institutional protocols, as well as differences between the two scans obtained for the same patient, can introduce heterogeneity that undermines reproducibility across centers and time points. Thus, we used non-contrast CT, which is more translatable to standard clinical workflows.
Regarding contouring variability, we followed established contouring guidelines used routinely at our institution for vulvar cancer. Moreover, for each patient, the same radiation oncology consultant delineated the primary GTV on both scans. However, we acknowledge the potential for single-observer bias and suggest that future studies incorporate multi-observer segmentation with formal inter-rater agreement metrics (e.g., Dice similarity coefficient, intraclass correlation) to further assess reproducibility. Discussion was edited accordingly.
Comment 4: Specify software versions and feature extraction parameters.
Response: In our revised manuscript, we report our detailed extraction setup in the “Methods”. Feature extraction used PyRadiomics’ RadiomicsFeatureExtractor with default parameters (no YAML overrides) on the Original CT image only, with 3D GTV masks (force2D=False) and no additional image filters. All feature classes were enabled.
Comment 5: Explain rationale for 107 features.
Response: We used the standard PyRadiomics default feature set to ensure a pre-specified, widely adopted, and reproducible feature space. In our environment, this yielded 107 features (First-order 18; Shape-3D 14; GLCM 24; GLRLM 16; GLSZM 16; GLDM 14; NGTDM 5), which we programmatically verified during extraction. We added these details to our revised manuscript, in the “Methods” section.
Comment 6: Justify p < 0.10 for univariate Cox screening.
Response: As sample size is small and given the exploratory intent of our study, a liberal screening threshold of p < 0.10 was adopted in the univariable Cox analysis to prevent premature exclusion of potentially informative features prior to multivariable analyses. This approach aligns with prior radiomics and exploratory modeling literature, which recommend inclusive thresholds at the pre-selection stage when subsequent penalized or stepwise methods (like LASSO, AIC) are applied to mitigate overfitting and control model complexity1–4. Such discussion was added to edited manuscript, “Methods” section.
- Van Griethuysen JJM, Fedorov A, Parmar C, Hosny A, Aucoin N, Narayan V, et al. Computational radiomics system to decode the radiographic phenotype. Cancer Res [Internet]. 2017 Nov 1 [cited 2025 Aug 7];77(21):e104–7. Available from: https://pubmed.ncbi.nlm.nih.gov/29092951/
- Parmar C, Grossmann P, Rietveld D, Rietbergen MM, Lambin P, Aerts HJWL. Radiomic Machine-Learning Classifiers for Prognostic Biomarkers of Head and Neck Cancer. Front Oncol [Internet]. 2015 [cited 2025 Oct 11];5(DEC). Available from: https://pubmed.ncbi.nlm.nih.gov/26697407/
- Aerts HJWL, Velazquez ER, Leijenaar RTH, Parmar C, Grossmann P, Cavalho S, et al. Decoding tumour phenotype by noninvasive imaging using a quantitative radiomics approach. Nat Commun [Internet]. 2014 Jun 3 [cited 2025 Aug 7];5. Available from: https://pubmed.ncbi.nlm.nih.gov/24892406/
- Zwanenburg A, Vallières M, Abdalah MA, Aerts HJWL, Andrearczyk V, Apte A, et al. The Image Biomarker Standardization Initiative: Standardized Quantitative Radiomics for High-Throughput Image-based Phenotyping. Radiology [Internet]. 2020 May 1 [cited 2025 Oct 11];295(2):328–38. Available from: https://pubmed.ncbi.nlm.nih.gov/32154773/
Comment 7: Address proportional hazards violations and time-varying covariates.
Response: Thank you for the important comment. As already incorporated in our methods and results sections, proportional-hazards assumptions were tested using Schoenfeld residuals (both global and covariate-specific). In the majority of models, both global and covariate-specific tests satisfied the PH assumption (p > 0.05). Minor time-dependent effects were observed for some Δ-features in the OS multivariable analysis. To account for these, we incorporated time-varying coefficients by including covariate × log(time) interaction terms. This adjustment improved model calibration without materially altering the direction or significance of the affected predictors. Although isolated proportionality deviations were observed, their magnitude was small, and their correction did not change the overall prognostic conclusions. This was also highlighted in the “Discussion” section in the edited manuscript.
Comment 8: Provide more patient details (comorbidities, prior treatments).
Response: In the revised manuscript, Table 1 was expanded with comorbidity and prior treatment data. Such data was also discussed in the edited “Results” section.
Comment 9: Univariate Cox results should be shown.
Response: Thank you for this comment. You can find the univariable Cox proportional-hazards analyses for all Δ-features and endpoints illustrated in Supplementary Table S1.
Comment 10: Justify AIC stepwise selection and risk of overfitting.
Response: Stepwise selection using Akaike’s Information Criterion (AIC) was utilized as it balances model goodness-of-fit and parsimony by penalizing excess parameters 5. While stepwise procedures can risk overfitting, this risk was mitigated through prior LASSO regularization, five-fold cross-validation, and 500-iteration bootstrap validation of concordance indices 6. Yet, overfitting is still possible given the small sample size, and we acknowledge that as a limitation of our study. These points were highlighted in edited manuscript.
- Akaike H. A New Look at the Statistical Model Identification. IEEE Trans Automat Contr. 1974;19(6):716–23.
- Harrell FE, Lee KL, Califf RM, Pryor DB, Rosati RA. Regression modelling strategies for improved prognostic prediction. Stat Med [Internet]. 1984 [cited 2025 Oct 12];3(2):143–52. Available from: https://pubmed.ncbi.nlm.nih.gov/6463451/
Comment 11: Interpret clinical significance of features.
Response: We further expanded the “Discussion” section. RLNU_norm, SVR, and texture heterogeneity were discussed, linking them to tumor morphology and treatment resistance.
Comment 12: Explicitly discuss selection bias, lack of validation, contouring subjectivity, and comparison with other imaging.
Response: Thank you for the suggestion. In the revised manuscript, limitations were further expanded to encompass the points suggested by this comment.
Comment 13: Emphasize need for multi-institutional validation.
Response: In the “Conclusion”, we recommend larger multi-institutional studies in order to validate the results we got in this study, preferably with integration with genomic/histopathologic data.
Round 2
Reviewer 1 Report
Comments and Suggestions for Authors
Thank you for revision. The paper is well-improved.
Reviewer 3 Report
Comments and Suggestions for Authors
Everything confirmed.